# Unmet clinical needs for COVID-19 tests in UK health and social care settings

**Sara Graziadio**[1]☯, **Samuel G. Urwin**[1]☯, **Paola Cocco**[2], **Massimo Micocci**[3], **Amanda Winter**[1], **Yaling Yang**[4], **D. Ashley Price**[1], **Mike Messenger**[2,5], **A. Joy Allen**[⬛][6‡]*, **Bethany Shinkins**[2,7‡], on behalf of the CONDOR Steering group¶

**1** NIHR Newcastle In Vitro Diagnostics Co-operative, Newcastle-upon-Tyne Hospitals Foundation Trust, Newcastle, United Kingdom, **2** NIHR Leeds In Vitro Diagnostics Co-operative, Leeds Teaching Hospitals NHS Trust, Leeds, United Kingdom, **3** NIHR London In Vitro Diagnostics Co-operative, Imperial College London, London, United Kingdom, **4** NIHR Community Healthcare MedTech and In Vitro Diagnostics Co-operative Nuffield Department of Primary Care Health Sciences, University of Oxford, Oxford, United Kingdom, **5** Leeds Centre for Personalised Health and Medicine, University of Leeds, Leeds, United Kingdom, **6** NIHR Newcastle In Vitro Diagnostics Co-operative, Newcastle University, Newcastle, United Kingdom, **7** Test Evaluation Group, Institute for Health Sciences, University of Leeds, Leeds, United Kingdom

☯ These authors contributed equally to this work.
‡ These authors are joint senior authors on this work. AJA and BS are also contributed equally to this work
¶ Membership of the CONDOR Steering group is provided in the acknowledgements
* Joy.Allen@newcastle.ac.uk

**Data Availability Statement:** Survey raw data file: DOI: 10.25405/data.ncl.13168274 Survey cleaned data file: DOI: 10.25405/data.ncl.13168277 Survey key: DOI: 10.25405/data.ncl.13168271

## Abstract

There is an urgent requirement to identify which clinical settings are in most need of COVID-19 tests and the priority role(s) for tests in these settings to accelerate the development of tests fit for purpose in health and social care across the UK. This study sought to identify and prioritize unmet clinical needs for COVID-19 tests across different settings within the UK health and social care sector via an online survey of health and social care professionals and policymakers. Four hundred and forty-seven responses were received between 22nd May and 15th June 2020. Hospitals and care homes were recognized as the settings with the greatest unmet clinical need for COVID-19 diagnostics, despite reporting more access to laboratory molecular testing than other settings. Hospital staff identified a need for diagnostic tests for symptomatic workers and patients. In contrast, care home staff expressed an urgency for screening at the front door to protect high-risk residents and limit transmission. The length of time to test result was considered a widespread problem with current testing across all settings. Rapid tests for staff were regarded as an area of need across general practice and dental settings alongside tests to limit antibiotics use.

## Introduction

Testing plays an integral role in the international response to the current coronavirus pandemic, facilitating national surveillance and guiding patient management. Testing strategies have been heavily scrutinized, and it is widely acknowledged that the tests currently available are far from ideal. For example, reverse transcription polymerase chain reaction (RT-PCR) based tests, the most commonly used laboratory-based tests for determining current infection, has a false negative rate of around 20% (depending on symptom duration, sampling technique, and site) [1].

 

**Funding:** This study is part of the CONDOR platform (condor-platform.org), which is funded by the UKRI, Asthma UK and the British Lung Foundation. SG, SGU, AW, DAP and AJA are supported by the National Institute for Health Research (NIHR) Newcastle In Vitro Diagnostics Co-operative (http://www.newcastle.mic.nihr.ac.uk/). MaM is supported by the NIHR London In Vitro Diagnostics Co-operative (https://london.ivd.nihr.ac.uk/). YY is supported by the NIHR NIHR Community Healthcare MedTech and In vitro Diagnostics Co-operative (https://www.community.healthcare.mic.nihr.ac.uk/home_). BS and MM are supported by the NIHR Leeds In Vitro Diagnostics Co-operative (https://www.leedsmic.nihr.ac.uk/). PC, MM and BS are supported by the 'Antimicrobial Resistance Cross Council Initiative' (Grant number MR/N029976/1), Funding Partners: The Biotechnology and Biological Sciences Research Council (https://bbsrc.ukri.org/), the Engineering and Physical Sciences Research Council (https://epsrc.ukri.org/), and the Medical Research Council (https://mrc.ukri.org/). This work is also supported by the Medical Research Foundation's National AMR Training Programme (https://www.medicalresearchfoundation.org.uk/projects/national-phd-training-programme-in-antimicrobial-resistance-research). The funders had no role in study design, data collection and analysis, decision to publish, or preparation of the manuscript

**Competing interests:** MM is a scientific advisor to the UK Department of Health and Social Care and a paid consultant for Cepheid Inc (unrelated to COVID-19) who receives research funding from Roche (unrelated to COVID-19). This does not alter our adherence to PLOS ONE policies on sharing data and materials. The remaining authors declare no other competing interests.

The industry response has been unprecedented; by the 17[th] of July 2020, 746 tests had been developed or are under development [2]. Although this extraordinarily fast development offers hope that the current testing shortfalls can be overcome, this creates a real challenge for diagnostic regulatory bodies. For many years, the conversion of diagnostic innovation from bench to clinical practice has been riddled with problems and is notoriously slow [3]. A rapid evaluation pipeline is required to ensure that tests offering real health benefits are integrated into practice within a timeframe that will support international efforts to curtail the transmission of SARS-CoV-2.

In March 2020, the World Health Organization (WHO) outlined a research roadmap recommending the development of Target Product Profiles (TPPs) to drive the research and innovation process around new diagnostic tests for COVID-19 [4]. A TPP is a document that summarises in advance the desirable and minimally acceptable specifications for a new test to address a well-defined clinical need. The overarching aim is to ensure that innovation efforts are focused on developing 'fit for purpose' tests [5,6]. The National Institute for Health and Care Excellence (NICE) have recently begun an economic modeling exercise to help inform TPP specifications for COVID-19 tests [7].

At the core of TPP development is the scoping and definition of unmet clinical needs [5]. Tests often have numerous roles in clinical practice across multiple clinical settings (e.g., primary and secondary care) and positions in a care pathway (e.g., to diagnose or to monitor disease) [8]. There is an urgent need to identify which health and social care settings are most in need of COVID-19 tests, and the priority role(s) for tests in these settings, particularly ahead of the winter season, where other respiratory viral-like illnesses will also be in circulation.

To this end, the National Institute for Health Research (NIHR) MedTech and In Vitro Diagnostics (IVD) Co-operatives (MICs), on behalf of the COVID-19 National DiagnOstic Research and Evaluation Platform (CONDOR) [9], designed an online survey at pace to gather opinion from health and social care professionals and policymakers on unmet clinical needs for COVID-19 tests in the UK. The primary objective of the survey was to determine which settings are in most need of a new COVID-19 test and determine the priority role(s) for new tests within each setting. The results will be used to help prioritise and refine TPP development, plan for winter allocation of tests, and inform economic modeling carried out by the NICE diagnostics assessment programme. Interim reports were shared with the Department of Health and Social Care (DHSC) and NICE on 29/05/2020 and 28/06/2020 to inform decisions around UK national testing policies.

## Materials and methods

This project was approved as a service evaluation (ID:10151) by the Newcastle Upon Tyne NHS Hospitals Foundation Trust. Consent to participate was obtained if the respondent read the information on the first page of the survey and clicked 'Consent to participate' (see S1 File). The development and reporting of the survey followed good practice guidelines published by Kelley et al. [10] and highlighted by the Enhancing the QUAlity and Transparency Of health Research (EQUATOR) network [11]. The survey questions were written based on an initial list of priority use cases developed by the DHSC COVID-19 Test Approvals Groups.

### Survey design, development, and dissemination

Targeted respondents of this survey were health and social care professionals working in the UK across a variety of settings, and policymakers.

The survey was voluntary and anonymized, with no personal, sensitive, or patient data collected. All respondents were asked to select which setting was most in need of a novel COVID-

19 test, and then health and social care professionals were asked to rate the importance of different use cases specific to the setting where they worked. The use cases described potential uses and roles of tests at specific points in the pathway; they were technology agnostic but described roles that could be fulfilled by molecular or serology tests. All respondents, apart from policymakers, were asked if diagnostic testing for COVID-19 was available in the setting in which they worked and, if so, they were asked which type of tests (e.g., molecular, serological) were available and whether they perceived any problems with the tests.

A first draft of the questionnaire was developed using Online surveys (formerly BOS) [12], with the support of two clinicians, and then the questionnaire was tested and revised by the wider authorship team to improve clarity and flow. The survey was then piloted by six hospital clinicians (one of whom worked closely with the ambulance service and another with care homes) and one dentist before its launch, with feedback incorporated into the survey's final version. Piloting the survey identified problems with wording and confirmed whether the survey functioned as intended in terms of access, navigation and submission [13,14]. Pilot data were not included in the final analysis to avoid contamination of results [14].

The survey was launched and disseminated via a variety of networks, including the NIHR Clinical Research Network (CRN) Coordinating Centre networks, clinical contacts of the NIHR MICs, as well as social media (limited to Twitter and LinkedIn, to our knowledge). A copy of the survey is available in the S1 File.

## Survey analysis

A setting was required to have at least 30 respondents in order to be included in the analysis, except for policymakers, where a smaller sample size was anticipated due to policymakers comprising a smaller proportion of the workforce compared to health and social care professionals.

Data processing and analysis were primarily conducted in the statistical programming language R [15], with qualitative free-text responses analyzed in Microsoft Excel. Where appropriate, combined heat maps and dendrograms were used to visualize results and identify similarities/differences between settings using the R function *'heatmap.2'* from the *'gplots'* package [16]. Hierarchical clustering was performed using the complete agglomeration method, and the Euclidean method was used to calculate the distances between the data.

Respondents selected whether they felt each use case in their specific setting was *'More Important'*, *'Important'* or *'Less Important'* relative to the others in the list. A scoring system was applied (More Important = 1, Important = 0, Less Important = -1), with the mean score and standard deviation calculated to summarize responses. The use cases were then ranked from the highest mean score to the lowest.

A framework for categorizing use cases was developed to facilitate the interpretation of results. Use cases were grouped in the survey based on their intended use: 'Screening' (to support isolation, personal protective equipment (PPE) use and cohorting decisions), 'Diagnostic' (to inform both infection control and treatment stratification), 'Prognostic' (escalation of care decisions) and 'Monitoring' (de-escalation of care and safe return to work) [17]. These were subsequently validated and independently re-grouped. In case of disagreement, a single arbiter resolved discrepancies. In the survey, the respondents could also suggest additional use cases via a free text field, if they perceived them to be missing from the provided list. Based on the previously defined framework, additional use cases were analyzed to identify responses similar to the provided list, then independently assigned to the 'Screening,' 'Diagnostic,' 'Prognostic,' and 'Monitoring' categories.

## Patient and public involvement

Seven members of the CONDOR [9] patient and public representative panel were invited to participate in a focus group discussion. The background, methods, and results of the survey were presented to the group, who were then invited to comment on these aspects. The participants provided verbal and written feedback during and after the meeting which has been incorporated into the discussion and will form the basis of a lay summary for the public to be published on the CONDOR website [9]. They agreed that the survey addressed important questions in relation to COVID-19 diagnostics. They raised the issues of geographical and ethnic representativeness of sampling, the potential omission of settings, and the importance of including patient experiences in future surveys. This included the impact of testing on their physical and mental health and the overall confidence of the public in COVID-19 testing.

## Results

The survey was launched at 4 pm on the 22nd of May 2020 and was open until 9 am on the 15th June 2020, receiving 447 completed responses (Table 1). It was not possible to estimate the number of potential respondents reached due to the variety of dissemination routes used. The distribution of survey responses over time, grouped by the setting in which the respondents worked, is presented in S1 Fig.

### Demographics of respondents

The majority of the respondents were from the North East of England (n = 217, 48.6%), with 411 respondents in total from England (92%) (Table 1).

The most common setting in which respondents worked was hospitals (n = 189, 42%), followed by primary dental care (n = 65, 15%) and general practice (n = 55, 12%) (Table 1). The hospice (n = 14, 3%), ambulance service (n = 9, 2%) and hospital-at-home settings (n = 3, 1%) received less than 30 respondents.

### Unmet clinical need

Settings which were deemed the highest priority were hospitals (Fig 1, 266 selections across all settings, 55.8% of respondents) and care homes (196 selections across all settings, 41.1% of respondents). This finding was apparent even when considering those who didn't select their own setting (the red bars in Fig 1). All other settings received less than 100 selections each. Twelve policymakers completed the survey. Most of them prioritized care homes (n = 4), followed by domestic residences (n = 3) and GP practices (n = 2), with hospitals, hospices, and the ambulance service selected by one respondent each. Care homes were generally prioritized by respondents working in other settings.

Use case ranking was explored in care homes with and without nursing independently, and the results were similar, as were those for primary and secondary dental care. We then pooled results from each of these two settings in order to increase the sample size for this and subsequent analyses.

The top three clinical needs for hospitals (Table 2) were all related to diagnosis to support infection control decisions for patients presenting to the hospital (1st), for workers (2nd), and for in-patients (3rd). For care homes, screening tests to support infection control decisions were the most important, with a test for those at admission and workers ranked first and second, respectively, and a test for patients potentially exposed ranked third. Respondents from GP and dental settings showed a similar focus on infection control measures for workers, identifying the two most important needs as testing to support screening and diagnosis of workers.

**Table 1. Characteristics of survey respondents.**

| Respondents, n | 447 | | | | |
|---|---|---|---|---|---|
| **Respondents' location, n (%)** | | | | | |
| North East | 217 | (48.6) | North West | 55 | (12.3) |
| South East | 33 | (7.4) | South West | 21 | (4.7) |
| East Midlands | 21 | (4.7) | Greater London | 20 | (4.5) |
| West Midlands | 19 | (4.3) | Scotland | 19 | (4.3) |
| Yorkshire and Humber | 16 | (3.6) | Wales | 12 | (2.7) |
| East of England | 9 | (2.0) | Northern Ireland | 5 | (1.1) |
| **Respondents' clinical setting, n (%)** | | | | | |
| Hospital* | 189 | (42.3) | Primary dental care | 65 | (14.5) |
| General practice | 55 | (12.3) | Prison | 30 | (6.7) |
| Secondary dental care[1] | 26 | (5.8) | Care home without nursing | 22 | (4.9) |
| Care home with nursing | 22 | (4.9) | Hospice | 14 | (3.1) |
| Health policy | 12 | (2.7) | Ambulance | 9 | (2.0) |
| Hospital-at-home | 3 | (0.7) | | | |
| ***Hospital sections where respondents in that setting worked, n (%)[2]** | | | | | |
| Outpatient clinic | 48 | (20.3) | General medicine ward | 28 | (11.9) |
| Other | 24 | (10.2) | Ophthalmology | 22 | (9.3) |
| Emergency Department | 19 | (8.1) | Intensive Care Unit | 19 | (8.1) |
| Oncology | 12 | (5.1) | Gastroenterology/Colorectal | 10 | (4.2) |
| General surgery ward | 9 | (3.8) | Respiratory | 8 | (3.4) |
| Laboratory | 7 | (3.0) | Theatres | 6 | (2.5) |
| Infectious diseases | 5 | (2.1) | Haematology | 5 | (2.1) |
| Rheumatology | 4 | (1.7) | Neurology | 4 | (1.7) |
| Renal | 3 | (1.3) | Endocrinology | 2 | (0.9) |
| Transplantation | 1 | (0.4) | | | |
| **Respondents' professional experience, n (%)** | | | | | |
| 10 or more years | 311 | (71.5) | Between 5 and 9 years | 71 | (16.3) |
| Between 1 and 4 years | 44 | (10.1) | Less than 1 year | 9 | (2.1) |
| **Patient groups that respondents worked with, n (%)[2]** | | | | | |
| Adults | 344 | (45.5) | Older people | 232 | (30.7) |
| Paediatrics | 143 | (18.9) | Neonates | 20 | (2.6) |
| None | 17 | (2.2) | | | |

[1]Secondary and community dental care;
[2]Multiple selection enabled question.

The third most important identified need was around monitoring for both settings. The dental setting showed a preference for a test to support safe return to work and the GP setting a test discharge patients into care homes safely. Respondents from the prison setting preferred a test to identify prisoners with COVID-19 at admission with symptoms (1st), and those with infection who are asymptomatic (2nd). The third most important identified need was to support workers in their self-isolation decisions when symptomatic; however, this had a much lower score than the first two ranked needs. No respondents across settings seemed to consider the few prognostic use cases proposed as being particularly important.

Some respondents suggested additional needs for clinical tests for COVID-19. While 32 (66.6%) of these could be mapped to existing use cases (the majority of which were for screening tests, n = 6 for patients, n = 14 for staff and n = 4 for community screening), 16 (33.3%)

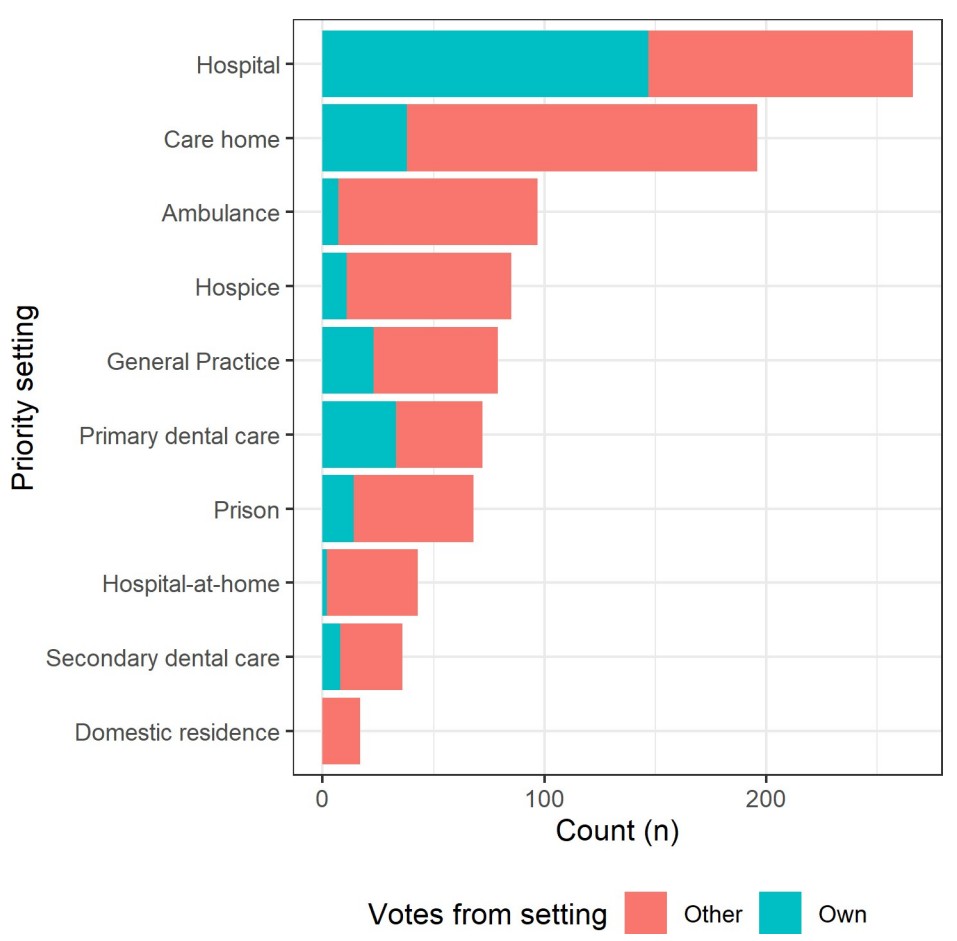

**Fig 1. Settings with the greatest perceived unmet clinical need for COVID-19 tests, showing votes from respondents for their own setting (blue shading) and other settings (red shading).**

were judged as new use cases by the authors (S1 and S2 Tables). Those that could not be mapped to existing use cases, highlighted the following additional needs: tests for 'track and trace', to detect immunity, to distinguish infective cases from non-active infections, to allow safe delivery of elective surgeries, and multiplex tests (i.e., panel tests that can detect multiple respiratory pathogens at the same time) for the winter season. Antibody tests (i.e., serology testing) were mentioned 12 times, with the majority of respondents suggesting that antibody tests would be useful to understand who had COVID-19, while some also recognized the uncertainty around antibody testing.

## Problems with medical tests and their availability

The settings where the respondents most frequently stated that COVID-19 tests were available (Fig 2) were hospitals (87%), prisons (85%), and care homes (78%). The settings where the respondents most frequently stated that no COVID-19 tests were available were dental care (86%) and GP (74%). In all settings where COVID-19 testing was reported as being available, molecular laboratory testing was the most common option. Serology point of care tests were only available in GP and hospital settings.

Respondents who stated that there were no COVID-19 tests available most frequently reported (Fig 3) that *'Potential spread of infection'* was the consequence of the lack of COVID-

**Table 2. Unmet clinical need priorities in each setting from respondents who worked in those settings, ranked by mean score from highest (most important clinical need) to lowest (least important clinical need).**

| Use cases | Hospital (n = 189) | | Care homes (n = 44) | | General Practice (n = 55) | | Dental setting (n = 91) | | Prison (n = 30) | |
|---|---|---|---|---|---|---|---|---|---|---|
| | Mean score | Rank | Mean score | Rank | Mean score | Rank | Mean score | Rank | Mean score | Rank |
| **SCREENING: to support isolation, PPE use and cohorting decisions** | | | | | | | | | | |
| A test for those on admission to prevent transmission to existing individuals | x | x | **0.93** | **1** | x | x | x | x | **0.80** | **2** |
| To determine whether an asymptomatic patient* has previously been infected with COVID-19 | 0.07 | 13 | 0.61 | 9 | 0.20 | 8 | 0.11 | 9 | 0.27 | 9 |
| A test for asymptomatic patients who have been potentially exposed | 0.42 | 6 | **0.70** | **3** | 0.22 | 7 | - | x | 0.53 | 5 |
| A test for asymptomatic patients to support safe attendance at routine appointments | x | x | x | x | x | x | 0.41 | 5 | x | x |
| A test for asymptomatic patients to support safe attendance at urgent appointments to guide appropriate use of PPE and aerosol generating procedures | x | x | x | x | x | x | 0.48 | 4 | x | x |
| A test for potentially exposed, asymptomatic workers to support self-isolation decisions | 0.43 | 5 | **0.77** | **2** | **0.67** | **1** | **0.81** | **1** | 0.60 | 4 |
| To identify who among patients presenting to clinical setting for reasons unrelated to COVID-19 can receive emergency treatment for non-COVID-19 conditions | 0.28 | 7 | x | x | x | x | x | x | x | x |
| To identify among patients presenting to clinical setting for reasons unrelated to COVID-19 who can receive routine treatment for non-COVID-19 conditions | 0.12 | 11 | x | x | 0.02 | 10 | - | x | x | x |
| **DIAGNOSTIC** | | | | | | | | | | |
| To determine whether a patient with flu-like symptoms has previously been infected with COVID-19 for infection control, patient triaging and management | 0.24 | 10 | 0.66 | 7 | 0.36 | 4 | 0.16 | x | 0.40 | 7 |
| A test for symptomatic patients presenting at the clinical setting to support infection control | **0.57** | **1** | 0.70 | 4 | 0.35 | 5 | x | x | **0.83** | **1** |
| A test for symptomatic patients to support safe attendance at urgent appointments to guide appropriate use of PPE and aerosol generating procedures | x | x | x | x | x | x | 0.33 | 6 | x | x |
| A guide for symptomatic patients to support safe attendance at routine appointments to guide appropriate use of PPE and aerosol generating procedures | x | x | x | x | x | x | 0.12 | 8 | x | x |
| A test for symptomatic care workers to support self-isolation decisions for infection control | **0.56** | **2** | 0.70 | 5 | **0.64** | **2** | **0.68** | **2** | **0.70** | **3** |
| A test for in-patients who develop new clinical features of COVID-19 during their stay to support infection control | **0.50** | **3** | | x | x | x | x | x | x | x |
| To confirm that a patient is currently infected with COVID-19 following initial/triage testing to inform treatment choices (e.g. antibiotics/antivirals) | 0.26 | 8 | 0.70 | 6 | 0.27 | 6 | x | x | 0.50 | 6 |
| **PROGNOSTIC: escalation of care decisions** | | | | | | | | | | |
| To identify who among patients with confirmed COVID-19 diagnosis could benefit from escalation of care | 0.08 | 12 | 0.61 | 8 | 0.04 | 9 | x | x | 0.30 | 8 |
| To identify among symptomatic individuals who could benefit from hospital admission | x | x | 0.57 | 10 | x | x | x | x | - | x |
| To determine whether a patient with flu-like symptoms has previously been infected with COVID-19 to support escalation of care advice | x | x | x | x | x | x | x | 7 | - | x |
| **MONITORING** | | | | | | | | | | |
| A test for care workers with a confirmed COVID-19 diagnosis to inform safe return to work | 0.47 | 4 | x | x | x | x | **0.63** | **3** | x | x |

(*Continued*)

**Table 2.** (Continued)

| Use cases | Hospital (n = 189) | | Care homes (n = 44) | | General Practice (n = 55) | | Dental setting (n = 91) | | Prison (n = 30) | |
|---|---|---|---|---|---|---|---|---|---|---|
| | Mean score | Rank | Mean score | Rank | Mean score | Rank | Mean score | Rank | Mean score | Rank |
| A test for in-patients with confirmed COVID-19 diagnosis to inform de-escalation of care/safe discharge (into care homes) and/or to move from side rooms | 0.24 | 9 | 0.41 | 11 | **0.40** | **3** | x | x | 0.10 | 10 |

Table key: 'PPE' = 'personal protective equipment', x indicates use cases that were not proposed for that setting. The three most important use cases identified by the respondents in each setting are indicated in bold text and shaded cells.

* Term 'patient' refers to residents and prisoners for care homes and prisons, respectively.

19 testing in the dental setting. '*Delays in administrating treatments to patient*' was the most frequently reported response. The hierarchical cluster analysis suggested that the perceived consequences of the lack of COVID-19 testing within the dental and GP settings were similar, where the misuse of antibiotics/antibiotic resistance was frequently reported. Within the prison setting, there were lower numbers reporting hospital and treatment-associated consequences.

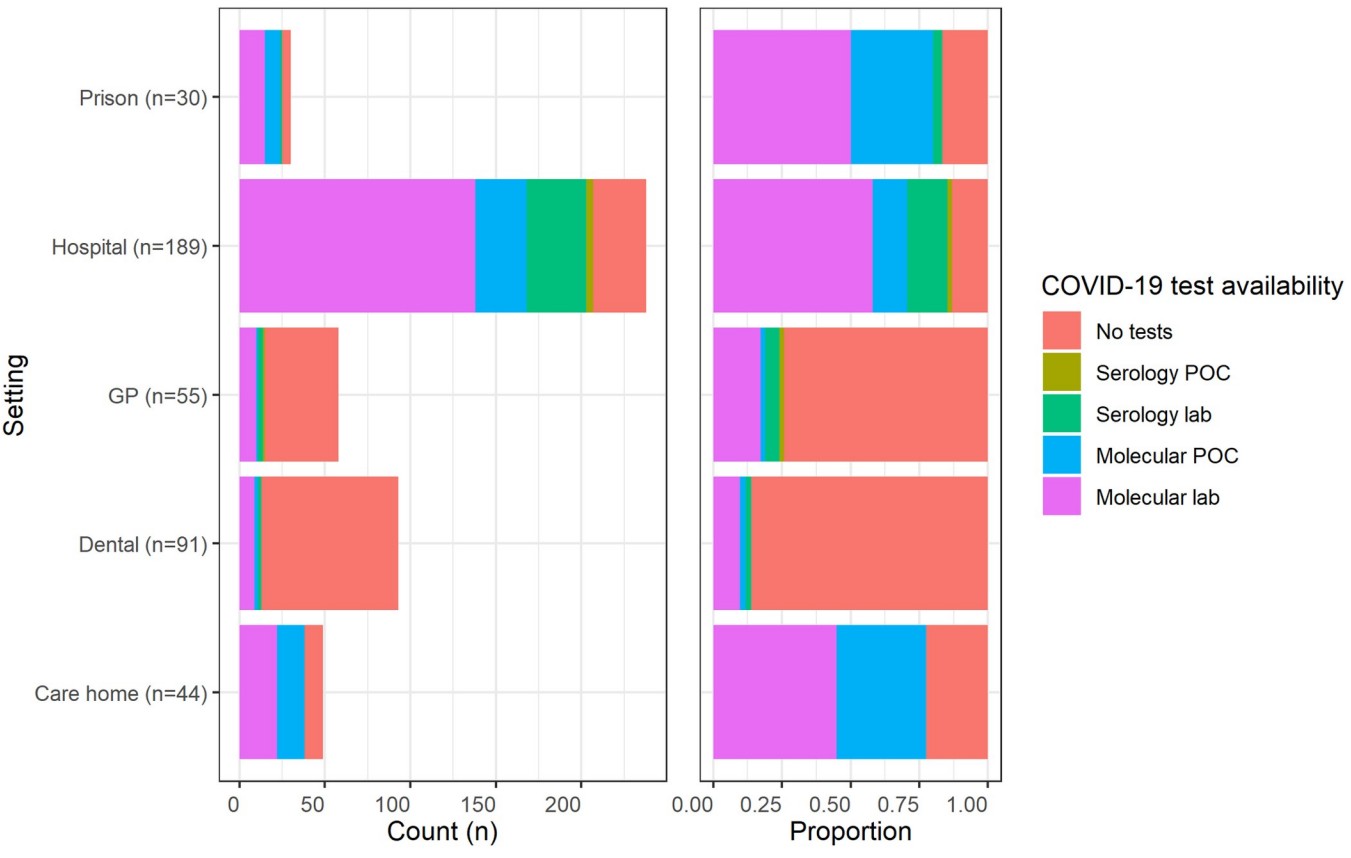

**Fig 2. The availability of COVID-19 tests in each setting as reported by respondents.** Primary and secondary dental settings were merged, in addition to care homes with and without nursing. The 'Hospital-at-home' (n = 3), hospices (n = 14) and ambulance setting (n = 9) were excluded due to the small number of overall respondents. Respondents were able to select multiple answers to this question. Plot key: 'GP' = 'General Practice'; 'POC' = 'Point of care'; 'lab' = 'Laboratory.

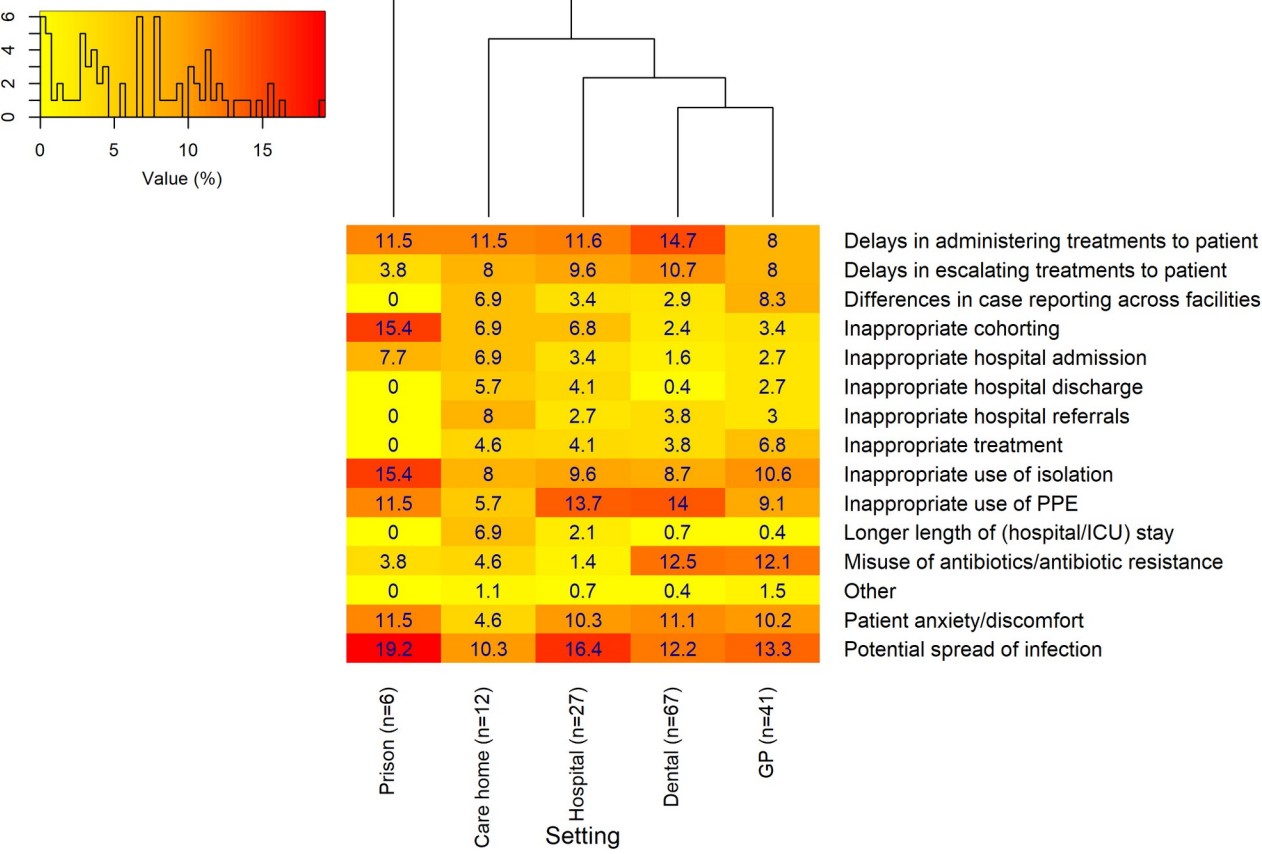

**Fig 3. Perceived consequences of no COVID-19 tests being available in each setting.** Primary and secondary dental settings were merged, in addition to care homes with and without nursing. The 'Hospital-at-home' (n = 3), hospice (n = 14) and ambulance setting (n = 9) were excluded due to the small number of overall respondents. A hierarchical cluster analysis indicates how similar/dissimilar the perceived consequences of no COVID-19 tests are between settings. The numbers of responses from individual settings for this particular question are displayed on the x axis (n = x). The histogram in the top left of the plot area shows the distribution of values, and provides a key to indicate the specific color shade of different values.

Across the majority of settings, most respondents stated that there were problems with testing in the setting where they worked (Fig 4). *'Long turnaround time'* for results of COVID-19 tests was reported most frequently as the most important problem across all settings, whereas *'Test usability'* and *'Too expensive'* were the least reported problem (Fig 5). There were similarities in the most important reported problems between hospital and GP respondents, with the second most frequent problem in the hospital and GP setting being the *'Poor ability to confirm if a patient does not have COVID-19'* (sensitivity), closely followed by the *'Poor ability to confirm if a patient has COVID-19* (specificity). Prison respondents reported a *'Lack of established protocols to inform decision making after positive/negative results'* as their most important problem. Responses from care homes and dental settings were also similar, with issues around the availability of equipment. *'Difficulty in obtaining a sample'* was reported more frequently for care homes than in other settings.

The most frequent consequence associated with these problems (Fig 6) was consistently reported as the *'Potential spread of infection'* along with *'Patient anxiety/discomfort,'* especially within the GP setting and care homes, and *'Inappropriate use of PPE'* in the dental setting. *'Repeat testing requirement'* in prisons and GP settings was also reported as a frequent consequence of these problems.

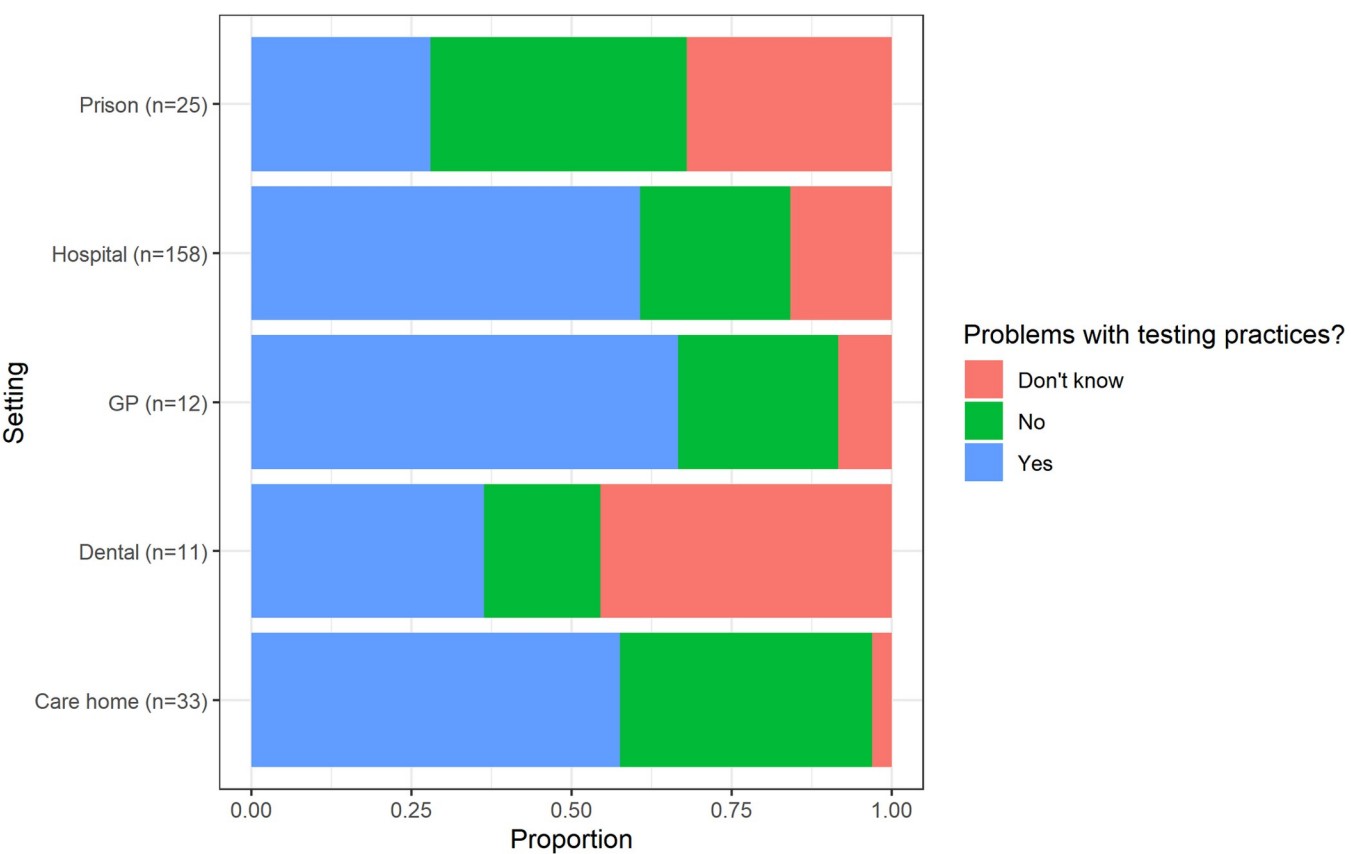

**Fig 4. Perception of problems associated with COVID-19 testing practices in each setting.** Primary and secondary dental settings were merged, in addition to care homes with and without nursing. The 'Hospital-at-home', hospice and ambulance setting were excluded due to the small number of overall respondents (n<15). Respondents selected a single 'Yes', 'No' or 'Don't know' answer. The number of responses from individual settings for this particular question is displayed on the y axis (n = x).

## Discussion

To our knowledge, this is the first survey of health and social care professionals and policy-makers to identify and prioritize unmet clinical needs for medical tests for COVID-19. We focused the survey within the UK health and social care system and covered a range of settings. Results of this survey informed the recent guidelines in the UK around testing for COVID-19 in various settings and are informing NICE early economic modeling. These results are important for planning policies for the next winter season when multiple respiratory conditions will be present and diagnostic tests will be crucial for the correct identification of COVID-19 patients.

### The unmet clinical need for testing

Hospital and care home settings (with and without nursing) were prioritized as being most in need of novel COVID-19 tests. Hospitals and care homes with nursing were reported as having greater access to testing compared to other settings, although care homes without nursing had minimal access to testing, highlighting a clear unmet clinical need.

The data suggested that hospitals were in most need of diagnostic tests (i.e., for symptomatic individuals), whereas respondents from care homes require tests to screen asymptomatic individuals. These differing roles were important for both patients and care workers across

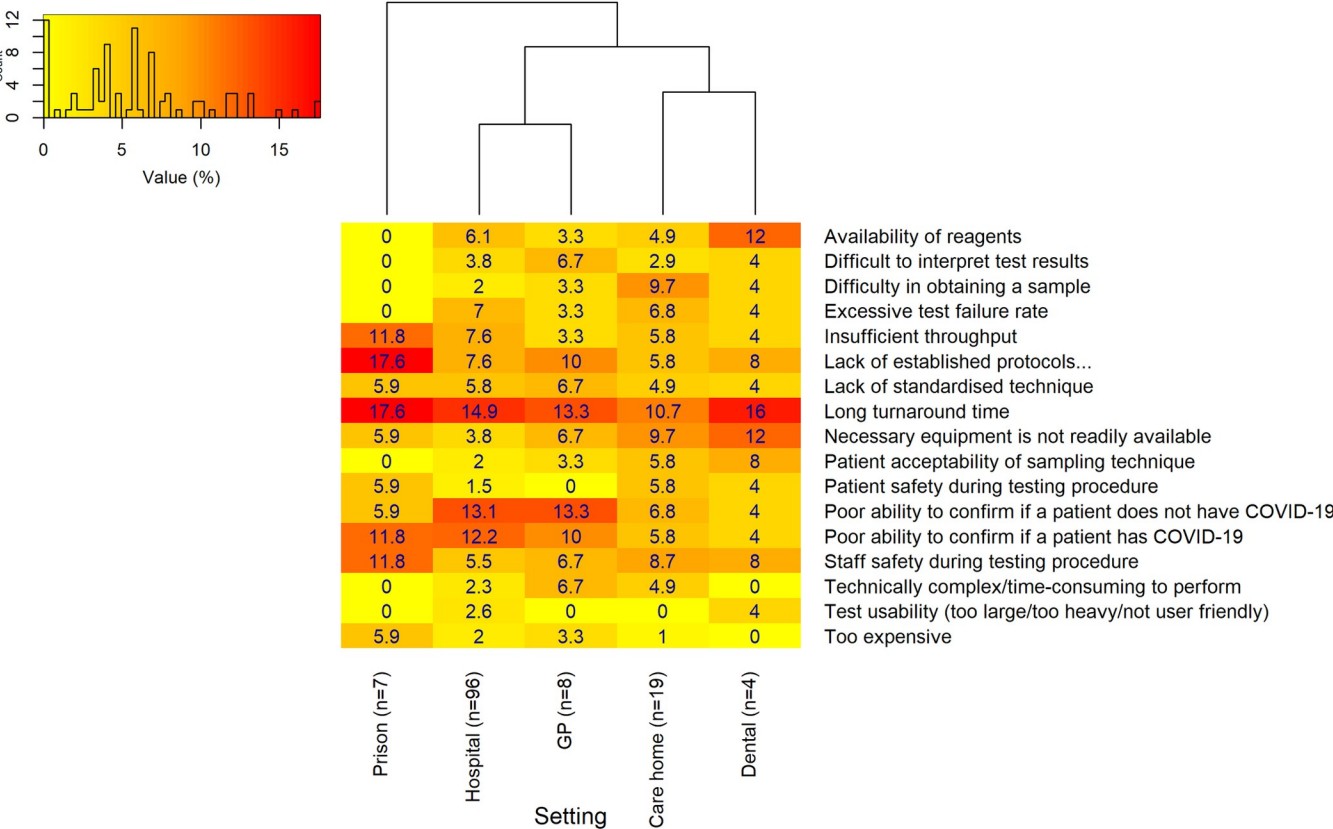

**Fig 5. Most important perceived problems associated with COVID-19 testing in each setting.** Primary and secondary dental settings were merged, in addition to care homes with and without nursing. The 'Hospital-at-home', hospice and ambulance setting (n<15) were excluded due to the small number of overall respondents. A hierarchical cluster analysis indicates how similar/dissimilar the perceived problems associated with COVID-19 testing are between settings. The numbers of responses from individual settings for this particular question are displayed on the x axis (n = x). The histogram in the top left of the plot area shows the distribution of values, and provides a key to indicate the specific color shade of different values.

these settings. The current testing policy in UK hospitals implemented on the 5th of June 2020, of testing every individual with potential COVID-19 symptoms [18], addresses this clinical need (the last hospital response to the survey was on the 3rd June 2020). The more extensive need for tests in care homes appears to reflect the increased risk for older people [19], where early identification of COVID-19 is extremely important for appropriate use of isolation (to decrease the risk of infection transmission) and treatment (to decrease the risk of hospital admission or death), the two major concerns associated with the lack of testing. Although it is important for care home residents to safely receive visitors, the overall risk to wider care home residents needs to be minimal, highlighting the importance of early identification of new cases and appropriate isolation use. The UK policy for care homes changed recently (27th July 2020 [20]), requiring weekly testing of staff and monthly testing of residents. This is in line with the clinical need identified through this survey but might pose a challenge in terms of test provision. In this context, the pooling of resident samples for testing might be a reasonable solution for screening in low prevalence conditions [21]. When a positive pool is detected, individual testing of residents and staff might follow.

Care homes reported issues with patient discomfort and difficulty in obtaining samples (more than others). Therefore, there is a need to develop tests that use sample types that are more readily obtained. Multiplex tests could also be useful in this population in the winter season, where the detection of multiple viruses is made possible through the collection of a single sample.

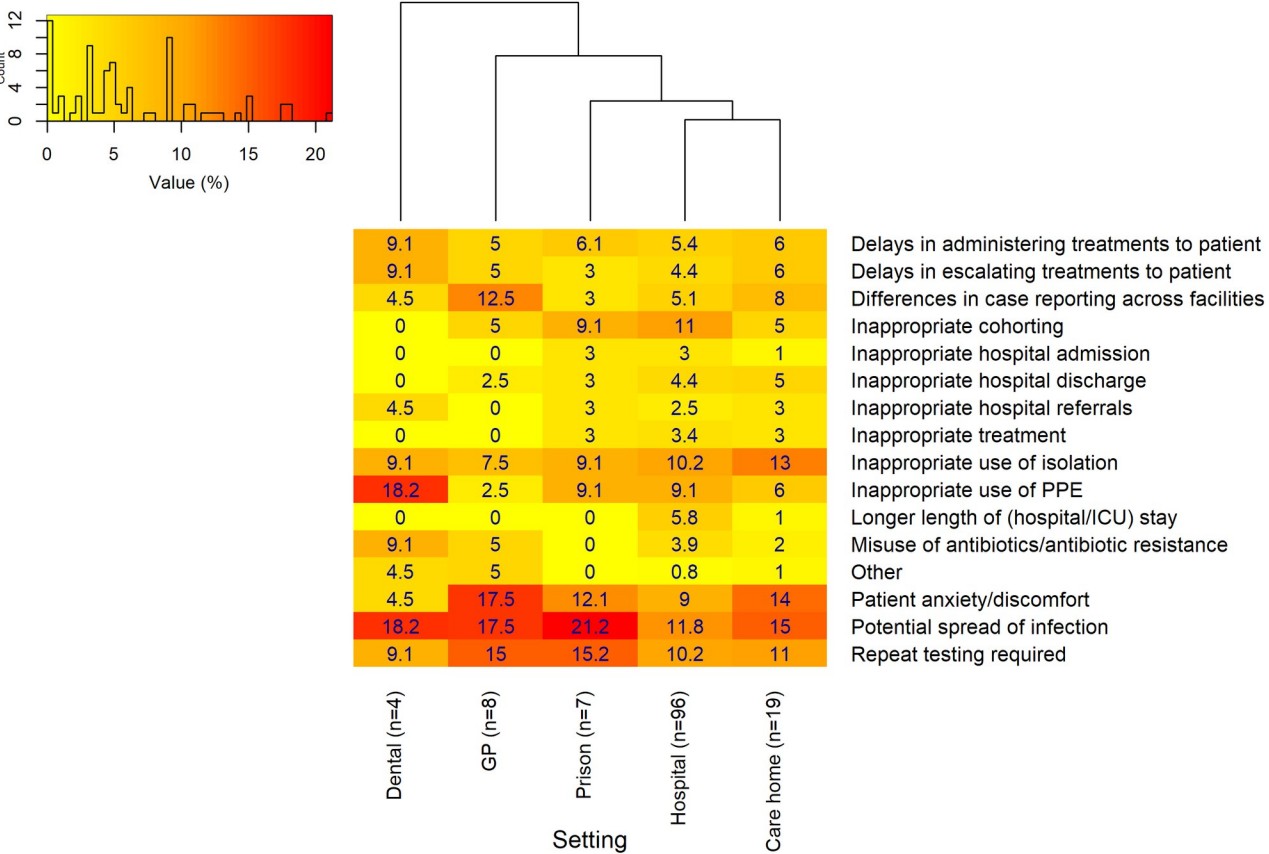

**Fig 6. Perceived consequences related to the problems associated with COVID-19 testing in each setting.** Primary and secondary dental settings were merged, in addition to care homes with and without nursing. The 'Hospital-at-home', hospice and ambulance setting (n<15) were excluded due to the small number of overall respondents. A hierarchical cluster analysis indicates how similar/dissimilar the perceived consequences related to the problems associated with COVID-19 testing are between settings. The numbers of responses from individual settings for this particular question are displayed on the x axis (n = x). The histogram in the top left of the plot area shows the distribution of values, and provides a key to indicate the specific color shade of different values.

The respondents from the prison setting indicated good availability of tests, with similar needs to the hospital setting for diagnosing and screening people at the front door to limit transmission of the virus to other prisoners. They did not have access to serology tests, but they seemed satisfied with the tests they had available, except for the long time to results and the lack of clear protocols to guide interpretation and actions following the results. Guidelines and care pathways should be developed in collaboration with health and social care professionals working in prisons to provide effective support during the winter season. This could help address health inequalities, which have been highlighted by the COVID-19 pandemic [22].

Interestingly, general practice and dental settings had similar needs which were mainly focused on identifying workers with COVID-19 (symptomatic or asymptomatic) who could transmit the disease to individuals in the community. Comparatively, these settings had limited access to COVID-19 tests and the main highlighted consequence of this was the inappropriate use of antibiotics. The most common new use cases stipulated by the respondents included tests that could support the Track and Trace program, surveillance of the general population as well as re-infection rates. Since policy decisions to limit or enhance lockdown measures are mainly based on these numbers [23], there is a need to increase testing capacity in these settings.

Across all settings, two main themes emerged. The first was the concern for the potential consequences of infection transmission, followed by the concerns around the long turnaround time for test results [24] to inform isolation decisions and PPE use. The role of treatment stratification for diagnostic tests might become more important in the future, once the clinical evaluation of treatments for COVID-19 are concluded (e.g. REMAP-CAP [25]).

## Limitations

There were limitations in the design of the survey. The survey was developed at pace, and as a result, could have benefitted from more comprehensive piloting, encompassing respondents from all targeted settings. The results had to be swiftly reported to inform government testing decisions, and therefore the survey was only open for a short period of time, which resulted in small sample sizes for some settings. Results for hospitals, care homes, general practices, and dental settings are likely to be more robust compared to other settings.

Rapid development and dissemination of the survey were required to capture the current picture of a fast-moving landscape. It may not have been optimal methodologically; however, it ensured an increased impact by informing decision makers. Furthermore, the number of misinterpreted use cases was low, ranging from 5 and 11% of the total number of respondents in each setting, and also the percentage of new use cases that were not included was low (0–11%). This seems to indicate good identification and clear wording of the use cases. The higher percentage of missed and unclear use cases reported by GP respondents (11%) may reflect the lack of piloting in that setting.

The survey was intended for response by health and social care professionals, although no questions were incorporated to validate inclusion criteria explicitly. Questions to identify high-risk staff were omitted.

Settings were targeted for dissemination at different times while the survey was open (see S1 Fig for the distribution of responses over time). The responses received may reflect changing clinical needs due to the fluctuating prevalence of COVID-19 and the different availability of tests during the pandemic. However, because the survey was only open for a short period of time covering three weeks at the tail end of the first wave of the pandemic in the UK, these effects are likely to be small.

The majority of the respondents worked in hospitals and in the North East of England, so results are less representative of other settings and other regions of the UK.

Finally, a limitation of qualitative research is that people who have a strong opinion on the subject are often the ones most likely to participate, potentially introducing selection bias. These limitations should be taken into account when inferring any conclusions from the data presented.

## Conclusions and next steps

In this first assessment of the clinical need for COVID-19 tests among experts in the UK, a long turnaround of results was the most commonly reported problem across all settings in May/June 2020, the consequence of which was identified as the potential transmission of infection. This would suggest a need for more rapid testing across all settings. Since it is unlikely that rapid near-patient testing will be available for the coming winter season to screen everyone, more work is needed to refine the higher priority groups and to understand the role of multiplex tests in the coming winter season.

From the current work, the groups with highest need for COVID-19 testing were identified as (in order):

- staff and patients with symptoms of COVID-19 in hospitals

- screening of staff, visitors and new residents in care homes

- staff in general practice and dental settings

- screening of new prisoners, visitors, existing prisoners and workers with symptoms of COVID-19.

These needs are most likely to be met by molecular tests, which are most accurate soon after the patient becomes infective [26]. Antibody tests seem less useful until the biology of the disease and its relationship with immunoglobulin A (IgA), G (IgG), and M (IgM) antibodies is clearer [27]. In care homes, quick and cheap tests with lower accuracy but frequent testing might be appropriate for screening, as suggested by modeling [28]. Alternatively, when prevalence is low, pooling of samples from residents could be a sensible approach for screening residents [21]. Rapid tests (under 10 minutes) that help distinguish viral from bacterial infections might be also useful in GP and dental setting to limit antibiotic resistance in the community. Despite this, more work is needed to fully characterize the unmet clinical need and optimize sample collection. It seems likely that multiplex tests will be useful in the winter season when other respiratory viruses circulate, particularly in care homes and in older populations of patients.

## Supporting information

**S1 Fig. Distribution of survey responses over time by setting.**
(TIF)

**S1 Table. Summary of the qualitative analysis of the new stipulated use cases proposed by the respondents in addition to the list designed by the authors.** Percentages in parenthesis refer to the total number of respondents for that setting.
(DOCX)

**S2 Table. Qualitative analysis of new stipulated use cases proposed by respondents.**
(DOCX)

**S1 File. COVID-19 unmet clinical needs survey.**
(DOCX)

## Acknowledgments

The authors would like to acknowledge the Department of Health and Social Care (DHSC) National Tests Advisory Group (NTAG) and the Viral Detection Test Advisory Group (VTAG) for sharing information vital to the development of the survey.

The authors would also like to acknowledge the professionals who piloted the survey: Dr. Malcolm Brodlie, Prof. Adam Gordon, Dr. Brendan Payne, Prof. Chris Price, Dr. Graham Walton, Prof. Mark Wilcox; who helped disseminate it: Prof. Paul Dark, Dr. Ewan Hunter, Dr. Lisa, Dr. Georgie McCann, Michelle O'Rourke and Prof. Rick Body; Marie Curie national network, The NIHR National Coordinating centre, the NIHR Clinical Research Network and Critical Care Network, the AHSN Network, the NHSA, the North East Local Dental Committees, the NHS Ambulance Service, the RCGP Research and Surveillance Centre, as well as the CONDOR steering group.

The authors gratefully acknowledge the CONDOR Patient and Public Involvement group (Andy Morgan, Chris Walker, Joan Bedlington, Julia Hamer-Hunt, Julia Roper, Margaret

Wilkinson and Graham Prestwich (Chair)) and Dr Rachel Dickinson for interesting discussions on the importance of the results. We would also like to acknowledge Dr. Clare Lendrem for helpful discussions and feedback on the survey and results and Prof. John Simpson for useful advice on the manuscript draft.

Additional members of the CONDOR Steering group include: Professor Richard Body, Dr Julian Braybrook, Professor Peter Buckle, Professor Paul Dark, Dr Kerrie Davis, Mrs Eloise Cook, Professor Adam Gordon, Mrs Anna Halstead, Professor Gail Hayward, Professor Dan Lasserson, Dr Andrew Lewington, Dr Brian Nicholson, Professor Rafael Perera-Salazar, Professor John Simpson, Dr Philip Turner, Mr Graham Prestwich, Dr Charles Reynard, Mrs Beverley Riley, Mrs Valerie Tate and Professor Mark Wilcox.

## Author Contributions

**Conceptualization:** Sara Graziadio, Mike Messenger, A. Joy Allen, Bethany Shinkins.

**Data curation:** Sara Graziadio, Samuel G. Urwin, Paola Cocco, Amanda Winter, A. Joy Allen, Bethany Shinkins.

**Formal analysis:** Sara Graziadio, Samuel G. Urwin, Paola Cocco, Massimo Micocci, Amanda Winter, Yaling Yang, Bethany Shinkins.

**Funding acquisition:** Sara Graziadio, A. Joy Allen.

**Investigation:** Sara Graziadio, Paola Cocco, Massimo Micocci, D. Ashley Price, A. Joy Allen, Bethany Shinkins.

**Methodology:** Sara Graziadio, Massimo Micocci, Mike Messenger, A. Joy Allen, Bethany Shinkins.

**Supervision:** Sara Graziadio, A. Joy Allen, Bethany Shinkins.

**Visualization:** Samuel G. Urwin.

**Writing – original draft:** Sara Graziadio, Samuel G. Urwin, Paola Cocco, A. Joy Allen, Bethany Shinkins.

**Writing – review & editing:** Sara Graziadio, Samuel G. Urwin, Paola Cocco, Massimo Micocci, Amanda Winter, Yaling Yang, D. Ashley Price, Mike Messenger, A. Joy Allen, Bethany Shinkins.

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
