## [Decision Letter · Decision Letter 0]

21 Oct 2020

PONE-D-20-28292

Unmet clinical needs for COVID-19 tests in UK health and social care settings

PLOS ONE

Dear Dr. Allen,

Thank you for submitting your manuscript to PLOS ONE. After careful consideration, we feel that it has merit but does not fully meet PLOS ONE’s publication criteria as it currently stands. Therefore, we invite you to submit a revised version of the manuscript that addresses the points raised during the review process.

We look forward to receiving your revised manuscript.

Kind regards,

Amit Sapra

Academic Editor

PLOS ONE

Journal Requirements:

2. Thank you for including your ethics statement:  "We obtained service evaluation approval from the Newcastle Upon Tyne NHS Foundation Trust to conduct the survey. The approval number is 10151.

The Human Research Authority online tool deemed that ethical approval was not required as there was no participant identifiable information obtained from the respondents.".   

Please provide additional details regarding participant consent. In the ethics statement in the Methods and online submission information, please ensure that you have specified (1) whether consent was informed and (2) what type you obtained (for instance, written or verbal, and if verbal, how it was documented and witnessed). If your study included minors, state whether you obtained consent from parents or guardians. If the need for consent was waived by the ethics committee, please include this information.

"I have read the journal's policy and the authors of this manuscript have the following competing interests:  MM is a scientific advisor to the UK Department of Health and Social Care and a paid consultant for Cepheid Inc (unrelated to COVID-19) who receives research funding from Roche (unrelated to COVID-19).

The remaining authors declare no other competing interests.  "

5. Please include a copy of Table 2 which you refer to in your text on page 8.

Reviewers' comments:

Reviewer's Responses to Questions

**Comments to the Author**

1. Is the manuscript technically sound, and do the data support the conclusions?

Reviewer #1: Yes

Reviewer #2: Yes

Reviewer #3: Yes

2. Has the statistical analysis been performed appropriately and rigorously? 

Reviewer #1: Yes

Reviewer #2: Yes

Reviewer #3: I Don't Know

3. Have the authors made all data underlying the findings in their manuscript fully available?

Reviewer #1: Yes

Reviewer #2: Yes

Reviewer #3: Yes

4. Is the manuscript presented in an intelligible fashion and written in standard English?

Reviewer #1: Yes

Reviewer #2: Yes

Reviewer #3: Yes

5. Review Comments to the Author

Reviewer #1: Good content but has lots of grammatical errors.

The article is well written and is relevant to the current health situation prevailing in the UK.

The study has good methodology. Authors have provided with tables with descriptive data.

Reviewer #2: Insightful research question, especially given the current times. It will help to inform future protocols in the healthcare setting especially with cold/flu season upon us. The limitations of the study are understandable given the short amount of time the study was conducted. The tables and figures were filled with information that was easy to interpret.

Reviewer #3: The following lines could be modified in the article:

"Testing strategies have been heavily scrutinised, and it is widely acknowledged that the tests currently available are far from ideal." Acknowledged by whom, please cite source.

"Although this level of innovation offers hope that the current testing shortfalls can be overcome, this creates a real challenge for diagnostic regulatory bodies." Please consider rephrasing, I could not understand what “this level of innovation" is referring to.

"The overarching aim is to ensuring that innovation efforts are focused on developing ‘fit for purpose’ tests", consider changing ensuring to ensure

6. PLOS authors have the option to publish the peer review history of their article (what does this mean?). If published, this will include your full peer review and any attached files.

Reviewer #1: **Yes: **Priyanka Bhandari

Reviewer #2: No

Reviewer #3: No

---

## [Author Response · Author response to Decision Letter 0]

22 Oct 2020

** Apologies, Table 2 was renamed Table 3 in error. This has been corrected ***

We thank the reviewer for their helpful comments. 

• We have addressed the grammatical concerns of reviewer 1 and made significant amends to the manuscript according to comments on the returned manuscript. 

• We have added an ethics statement plus further information on the consenting process. 

• Apologies that you were unable to view Table 2. It is included at the end of the manuscript which has been reuploaded. 

• We have updated the format of our manuscript to fit within the PLOS One guidelines which have been shared 

• We have also updated our Supplementary Material to ‘Supporting Material’ and uploaded copies with and without tracked changes (‘SupportingInformation’ and ‘SupportingInformation with tracked changes’ respectively).

• Our updated conflict of interest statement is ‘MM is a scientific advisor to the UK Department of Health and Social Care and a paid consultant for Cepheid Inc (unrelated to COVID-19) who receives research funding from Roche (unrelated to COVID-19). This does not alter our adherence to PLOS ONE policies on sharing data and materials.

The remaining authors declare no other competing interests.

---

## [Editor Report · Decision Letter 1]

28 Oct 2020

Unmet clinical needs for COVID-19 tests in UK health and social care settings

PONE-D-20-28292R1

Dear Dr. Allen,

We’re pleased to inform you that your manuscript has been judged scientifically suitable for publication and will be formally accepted for publication once it meets all outstanding technical requirements.

Kind regards,

Amit Sapra

Academic Editor

PLOS ONE
---

## [Editor Report · Acceptance letter]

4 Nov 2020

PONE-D-20-28292R1 

Unmet clinical needs for COVID-19 tests in UK health and social care settings 

Dear Dr. Allen:

I'm pleased to inform you that your manuscript has been deemed suitable for publication in PLOS ONE. Congratulations! Your manuscript is now with our production department. 

Kind regards, 

on behalf of

Dr. Amit Sapra 

Academic Editor

PLOS ONE